# Using triangulation to evaluate findings from random-intercept cross-lagged panel models: An application with data on curiosity and creativity

**Kimmo Sorjonen** *, **Bo Melin**

Department of Clinical Neuroscience, Karolinska Institutet, Stockholm, Sweden

* kimmo.sorjonen@ki.se

## Abstract

In a recent study, researchers found cross-lagged effects between curiosity and creativity in an analysis with the random-intercept cross-lagged panel model (RI-CLPM) and concluded that curiosity and creativity mutually reinforce each other. However, it is known that the RI-CLPM can give biased results. Here, we used triangulation and analyzed the same data ($N = 400$) with additional models, including a latent change score model (LCSM) and multilevel regression analyses of person-mean centered scores. Only results from the original RI-CLPM were consistent with the conclusion of mutually reinforcing effects between curiosity and creativity while results from the other models contradicted this conclusion. Moreover, the model of spurious longitudinal associations (MoSLA) suggested that data might have been generated without any direct effects between curiosity and creativity. An aggregation of available evidence made us conclude that longitudinal associations between curiosity and creativity in the present data probably were spurious, possibly due to confounding by a trait common to curiosity and creativity and common auto-correlated state factors with effects on curiosity and creativity measured at the same occasion. The present study, and the available analytic script, can be used as a model/tutorial by researchers wishing to scrutinize results from the RI-CLPM.

## Introduction

Triangulation and trilateration are methods for establishing locations. For example, if we know that a sought location in Fig 1 is 10 km from A, it is possible that P1 is the location. However, if we receive the additional information that the sought location is 7 km from B, we now know that P1 cannot be the location, as the location must be either P2 or P3. With a third piece of information that the sought location is 6 km from C, we can establish that P3 is the location (in a two-dimensional space).

In a wider meaning, triangulation is a term used for methods where information is gathered through several different channels and a position is adopted or a decision

**Data availability statement:** The analytic script, which also downloads the used data, is available at the Open Science Framework at https://osf.io/rk4ne/.

**Funding:** The author(s) received no specific funding for this work.

**Competing interests:** The authors have declared that no competing interests exist.

reached based on an aggregation of available information. The more information that speaks for a specific position, and the less that speaks against it, the more confident one can be in adopting the position. Triangulation in research, e.g., by comparing results across methods with different sources and direction of potential bias, has been endorsed [1,2]. However, according to our experience, triangulation is rare in psychological research. Instead, the standard procedure appears to be to have a pre-formulated hypothesis and if the hypothesis is at least partly consistent with data at hand it is concluded, explicitly or implicitly, that the hypothesis is correct. This procedure ignores the fact that "is consistent with" is not the same as "proves". For example, a hypothesis that P1 in Fig 1 is the sought location is consistent with the information that the location is 10 km from A, but this does not prove that P1 is the location because there are a multitude of other locations that also are 10 km from A.

In agreement with Munafó and colleagues [1,2], we believe that causal conclusions based on analyses of correlational (i.e., non-experimental) data could be evaluated through triangulation. For example, Ma and Wei [3] analyzed data on curiosity and creativity with a random-intercept cross-lagged panel model (RI-CLPM) and concluded, based on statistically significant cross-lagged effects, that curiosity and creativity mutually reinforce (i.e., cause) each other. However, it is known that RI-CLPM, similarly as the traditional cross-lagged panel model (CLPM), can give biased results [4–7]. For example, Sorjonen et al. [8] showed that if longitudinal measures of two constructs X and Y are affected by common auto-correlated state factors, RI-CLPM will tend to indicate statistically significant, but spurious, cross-lagged effects. Consequently, although statistically significant cross-lagged effects in RI-CLPM are consistent with causal effects between X and Y, they do not prove causality any more than the fact that the sought location in Fig 1 is 10 km from A proves that P1 is the location.

One possible type of triangulation would be to use different measures of the predictors and outcomes of interest. If results from analyses of different measures converge, conclusions can be drawn with increased confidence. If, on the other hand, results diverge, caution is advised and confident conclusions should probably be avoided. Another type of triangulation could be to analyze models where effects are predicted to have different signs if a certain interpretation of findings is correct. For example, let us assume that pouring water into a cup has an increasing effect on its total weight, i.e., including its content. If we pour different amounts of water into cups with the same initial weight, those cups that receive most water should have the highest subsequent weight. This corresponds to a positive effect of the amount of poured water on subsequent weight when adjusting for initial weight. For example, with the data in Fig 2A, the effect of amount of poured water (milliliters) on weight at T2 (grams) when adjusting for weight at T1 (i.e., conditioning on the same weight at T1) would equal $b = 1$.

However, if the effect is truly increasing, if we pour different amounts of water into cups that then have the same subsequent weight, those cups that receive most water should have had the lowest initial weight. High initial weight would have compensated for receiving little water and allowed the cup to have the same subsequent weight as

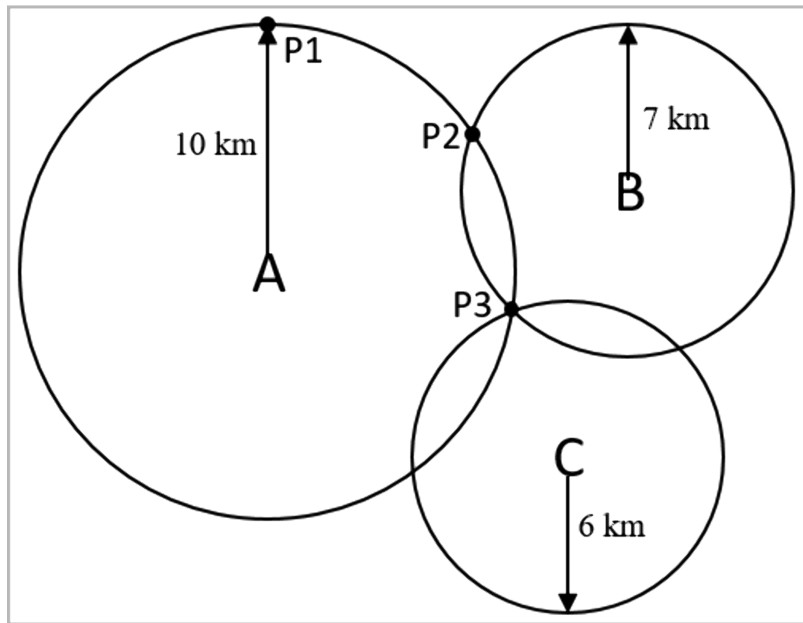

**Fig 1. Trilateration.** An example of trilateration used for establishing a location.

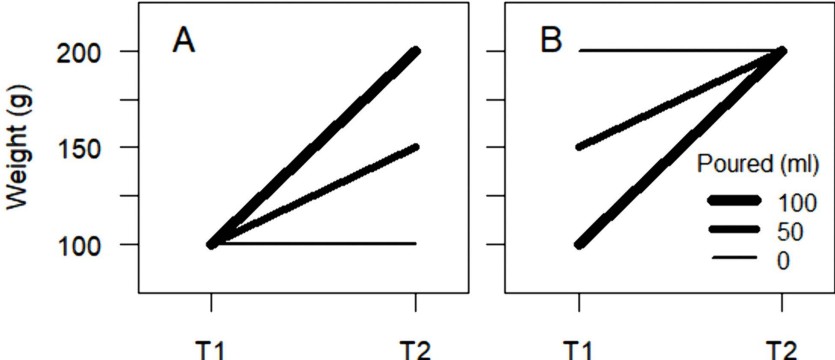

**Fig 2. Effect of water. (A)** Positive effect ($b=1$) of amount of poured water on the weight of cups at time 2 (T2) when conditioning on the same weight at time 1 (T1); **(B)** Negative effect ($b=-1$) of amount of poured water on the weight of cups at T1 when conditioning on the same weight at T2.

cups that received more water. This corresponds to a negative effect of the amount of poured water on initial weight when adjusting for subsequent weight. With the data in Fig 2B, the effect of amount of poured water (milliliters) on weight at T1 (grams) when adjusting for weight at T2 (i.e., conditioning on the same weight at T2) would equal $b=-1$.

Moreover, if the effect is truly increasing, we should see the most positive subsequent-initial weight difference for cups that receive most water, i.e., a positive crude effect of the amount of poured water on the subsequent-initial weight difference. With the data in Fig 2A and 2B, the effect of amount of poured water (milliliters) on the weight at T2 - weight at T1 difference (grams) would equal $b=1$.

Similarly as in the example with water above, if, as concluded by Ma and Wei [3], initial curiosity has an increasing effect on creativity, initial curiosity should have a positive effect on subsequent creativity when adjusting for initial creativity, a negative effect on initial creativity when adjusting for subsequent creativity, and a positive effect on the subsequent-initial creativity difference.

We have previously used triangulation, i.e., analyzed data with alternative models, to challenge conclusions based on findings from analyses with the traditional CLPM [e.g., 9–11]. Moreover, in a recent study we used triangulation to scrutinize and challenge conclusions of causal effects of academic self-concept (i.e., self-perceived academic competence) on academic achievement based on findings from analyses with the RI-CLPM [12]. The objective of the present study was to further illustrate how triangulation can be used to evaluate findings from RI-CLPM through reanalyses of data used by Ma and Wei [3]. If Ma and Weis' conclusion that curiosity has an increasing effect on creativity was correct, we should see the combination of positive and negative effects predicted above. If we do not see the predicted combination of positive and negative effects, the conclusion by Ma and Wei would appear premature. Based on findings by Sorjonen et al. [8], we also fitted an alternative model to the data on curiosity and creativity. In this model, longitudinal measures of the two constructs were affected, in addition to trait factors, by common auto-correlated state factors but they did not have any direct effects on each other. If this model fits data, it will be a further indication that the findings by Ma and Wei may have been spurious and that their conclusion can be challenged.

## Method

### Ethics statement

This study involves secondary data analysis of data that were used in compliance with the primary study guidelines. We have depended on those who collected the data to obtain ethics committee approval and informed consent. According to the authors of the primary study, the data collection was approved by the primary study's first author's affiliated institute and complied with the ethical guidelines of the American Psychological Association (APA) [3].

### Data

We refer to Ma and Wei [3] for more comprehensive information on the study sample, used instruments, study procedure, etc. In short, Ma and Wei collected data from employees at three firms in Hongkong, China ($N=400$). At three occasions (three months apart), the employees answered seven items measuring their work-related curiosity while their supervisors answered four items measuring the employees' creativity. Ma and Wei have made their data available at the Open Science Framework at https://osf.io/d48xa/.

### Analyses

Data were analyzed with seven different models (regression equations for effects included in these models can be found in the document "Supplementary_Equations" available at https://osf.io/rk4ne/):

1. Similarly as Ma and Wei [3], we analyzed data with a random-intercept cross-lagged panel model (RI-CLPM). RI-CLPM is an extension of the traditional cross-lagged panel model (CLPM), where longitudinally measured scores on two variables are regressed on two latent variables (the random intercepts) corresponding to individuals' general trait-like levels on the two constructs, respectively. Then, auto-regressive and cross-lagged effects are estimated between residuals of the scores not accounted for by the trait-like levels. In this way, effects are presumably estimated within individuals rather than between individuals as in the traditional CLPM [13,14]. Within-individual cross-lagged effects are assumed to be less biased and, consequently, better estimates of increasing/decreasing effects than cross-lagged effects in traditional CLPM [15].

2. We fitted an alternative RI-CLPM to data, where initial within-individual residuals of creativity were regressed on initial within-individual residuals of curiosity as well as subsequent within-individual residuals of creativity. Similarly as the

amount of poured water is expected to have a negative effect on the initial weight of cups when adjusting for their subsequent weight (see the Introduction and Fig 2B), with a truly increasing effect, initial within-individual residuals of curiosity were expected to have a negative effect on initial within-individual residuals of creativity when adjusting for subsequent within-individual residuals of creativity.

3. We fitted a latent-change score model (LCSM) to data. In LCSM, subsequent scores on variables of interest are regressed on initial scores as well as on latent change scores, with both regression effects set to 1. In this way, the subsequent score is defined as initial score + change and, consequently, the latent change score as subsequent score – initial score. Then, the latent change score can be regressed on other variables, e.g., an initial score on some other variable of interest, in order to evaluate effects on change [16–18]. If curiosity and creativity have true reinforcing reciprocal effects on each other, we should see a positive effect of initial curiosity on subsequent latent change in creativity and vice versa.

4. We estimated person-mean centered scores on curiosity and creativity as alternative measures of within-individual levels to the within-individual residuals estimated in RI-CLPM. On person-mean centered scores, values 1 and −1 mean that on those occasions the individual was one scale point above and below hers/his average score across measurements, respectively. Using multilevel regression analysis, we estimated the effect of initial person-mean centered curiosity on subsequent person-mean centered creativity when adjusting for initial person-mean centered creativity. With a true within-individual increasing effect of curiosity on creativity, this effect was expected to be positive. This analysis corresponded to the original RI-CLPM (model 1), but with a different way of measuring within-individual scores and using multilevel regression instead of structural equation modeling.

5. Corresponding to model 2, we used multilevel regression to estimate the effect of initial person-mean centered curiosity on initial person-mean centered creativity when adjusting for subsequent person-mean centered creativity. Similarly as in model 2, with a true increasing effect, this effect was expected to be negative.

6. Corresponding to model 3 (the LCSM), we used multilevel regression to estimate the effect of initial person-mean centered curiosity on the subsequent-initial creativity difference. Similarly as in model 3, with a true increasing effect, this effect was expected to be positive.

7. We fitted a model where the three measures of curiosity and creativity were affected both by trait-like factors (random intercepts) and by common auto-correlated state factors. We call this the model of spurious longitudinal associations (MoSLA). If the MoSLA exhibits acceptable fit to data, it would indicate that data may have been generated by a model with no direct effects between curiosity and creativity and that cross-lagged effects in the RI-CLPM, consequently, may have been spurious.

Analyzing models 2 and 5 were in line with proposals that time-reversed analyses may be used to identify potential statistical artifacts [19,20]. Previous analyses of data including an unquestionable causal effect (of adding stones in a container on the total weight of the container) found consistent effects in models 1–6 (i.e., effects in line with the predictions outlined above), suggesting that these models may be used to discriminate between true causal and spurious effects [21]. Model 7 (the MoSLA) was used in two recent studies and suggested that longitudinal data on academic self-concept and achievement [12] and on executive deficits and psychopathology [22], respectively, may have been generated without any direct effects between these constructs and, consequently, that previous causal claims could be challenged.

Analyses were conducted with R 4.3.1 statistical software [23] employing the osfr [24], lavaan [25], and lmerTest [26] packages. The analytic script, which also downloads the used data, is available at the Open Science Framework at https://osf.io/rk4ne/.

## Results

Results from models 1–3 and 7 are presented in Fig 3, with all models exhibiting good fit to data (full output for all models, including standard errors, p-values, etc., can be found in the document "Supplementary_Output" available at https://osf.io/rk4ne/). The RI-CLPM confirmed Ma and Wei's [3] finding of an effect of initial within-individual residual of curiosity on subsequent within-individual residual of creativity when adjusting for initial within-individual residual of creativity (e.g.,

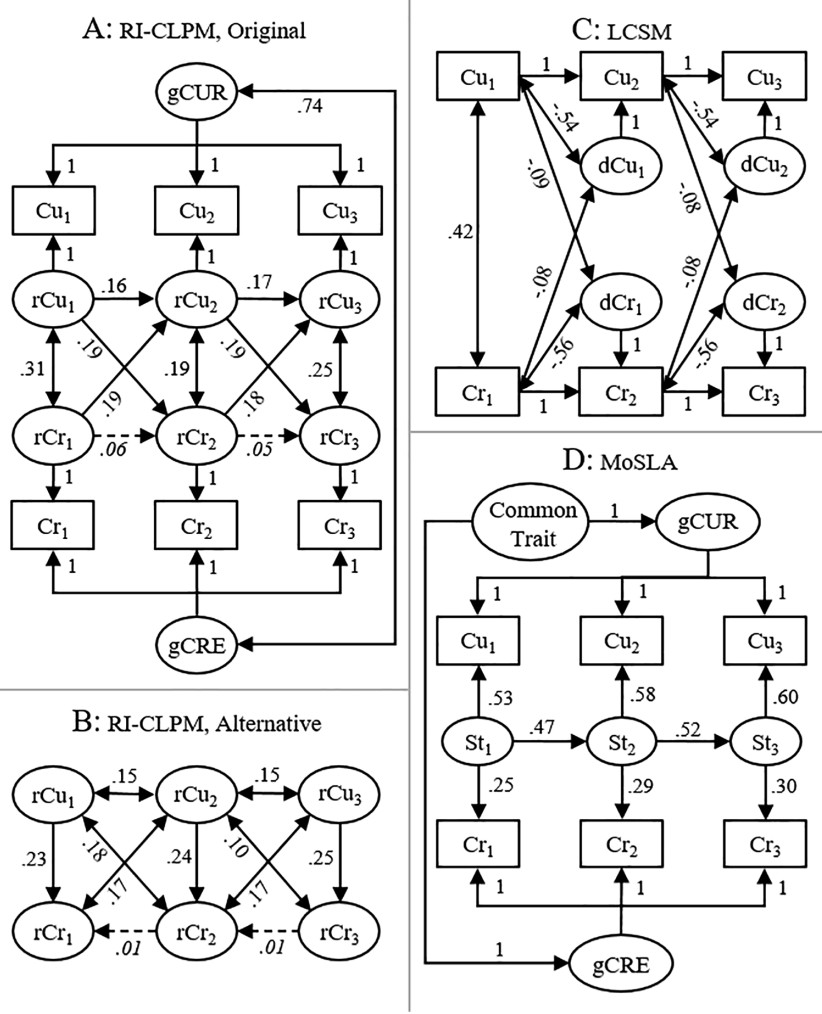

**Fig 3. The models. (A)** Original random-intercept cross-lagged panel model (RI-CLPM), with initial within-individual residuals of curiosity predicting subsequent within-individual residuals of creativity when adjusting for initial within-individual residuals of creativity and vice versa. The model exhibited good fit to data ($\chi^2 = 2.10$, DF = 5, $p = 0.835$, CFI = 1.00, TLI = 1.02, RMSEA = 0.00 [90% CI: 0.00; 0.04]); **(B)** Alternative RI-CLPM, with initial within-individual residuals of curiosity predicting initial within-individual residuals of creativity when adjusting for subsequent within-individual residuals of creativity. Only the associations between within-individual residuals are shown. The model exhibited good fit to data ($\chi^2 = 2.60$, DF = 5, $p = 0.761$, CFI = 1.00, TLI = 1.02, RMSEA = 0.00 [90% CI: 0.00; 0.05]); **(C)** Latent change score model (LCSM), with initial curiosity predicting subsequent latent change in creativity and vice versa. The model exhibited good fit to data ($\chi^2 = 25.7$, DF = 12, $p = 0.012$, CFI = 0.97, TLI = 0.96, RMSEA = 0.05 [90% CI: 0.02; 0.08]); **(D)** Model of spurious longitudinal associations (MoSLA), where scores on curiosity and creativity are affected by general trait-like factors as well as auto-correlated state factors. The model exhibited good fit to data ($\chi^2 = 14.9$, DF = 9, $p = 0.094$, CFI = 0.99, TLI = 0.98, RMSEA = 0.04 [90% CI: 0.00; 0.08]). Note: 1,2, and 3 = occasion 1, 2, and 3, respectively; Cu = curiosity; Cr = creativity; gCUR/gCRE = general, trait-like, curiosity and creativity, respectively; rCu/rCr = within-individual residual of curiosity and creativity, respectively; dCu/dCr = latent change in curiosity and creativity, respectively; St = state factor. Standardized coefficients (except those set to 1). All coefficients stronger than 0.07 (i.e., solid arrows) are statistically significant ($p < 0.05$).

β = 0.187 [0.040; 0.333], *p* = 0.012) and vice versa (e.g., β = 0.192 [0.066; 0.318], *p* = 0.003) (model 1, Fig 3A). This corresponds to the amount of poured water having, as expected, a positive effect on the weight of cups at T2 when conditioning on the same weight at T1 (Fig 2A).

However, contrary to expectations by a hypothesis of truly increasing effects, initial within-individual residual of curiosity had a positive effect on initial within-individual residual of creativity when adjusting for subsequent within-individual residual of creativity (e.g., β = 0.232 [0.107; 0.358], *p* < 0.001) (model 2, Fig 3B). This means that among individuals with the same subsequent within-individual residual of creativity (e.g., 0), those with high initial within-individual residual of curiosity (e.g., 1) were predicted to have had higher initial within-individual residual of creativity (1 × 0.232 = 0.232) compared with those with lower (e.g., −1) initial within-individual residual of curiosity (−1 × 0.232 = −0.232) and, consequently, to have decreased more in within-individual residual of creativity between the measurements (0 − 0.232 = −0.232 vs. 0 − (−0.232) = 0.232). This positive effect suggests that low initial within-individual residual of curiosity had compensated for low initial within-individual residual of creativity and allowed individuals to reach the same subsequent within-individual residual of creativity as those with higher initial within-individual residual of creativity and higher initial within-individual residual of curiosity. This positive effect would correspond to the amount of poured water having, contrary to expectations illustrated in Fig 2B, a positive effect on the weight of cups at T1 when conditioning on the same weight at T2.

Also contrary to predictions by a hypothesis of truly increasing effects, the effect of initial curiosity on subsequent latent change in creativity was negative (e.g., β = −0.085 [−0.157; −0.012], *p* = 0.023) and vice versa (e.g., β = −0.077 [−0.147; −0.007], *p* = 0.032) (model 3, Fig 3C). These negative effects would correspond to the amount of poured water having, contrary to expectations, a negative effect on the weight at T2 - weight at T1 difference of cups (Fig 2).

The multilevel regression analyses (models 4−6) found no statistically significant effect of initial person-mean centered curiosity on subsequent person-mean centered creativity when adjusting for initial person-mean centered creativity (model 4, β = −0.013 [−0.091; 0.063], *p* = 0.734). Moreover, contrary to predictions by a hypothesis of truly increasing effects, initial person-mean centered curiosity had a positive effect on initial person-mean centered creativity when adjusting for subsequent person-mean centered creativity (model 5, β = 0.083 [0.009; 0.158], *p* = 0.029) and a negative effect on the subsequent – initial creativity difference (model 6, β = −0.118 [−0.214; −0.023], *p* = 0.016). Furthermore, the MoSLA (model 7), where measures of curiosity and creativity were affected by trait-like factors (random intercepts) as well as common auto-correlated state factors but without any direct effects on each other, exhibited good fit to data (Fig 3D).

In models 2 and 4–6, we limited analyses to models with curiosity as the predictor and creativity as the outcome. Corresponding models with creativity as the predictor and curiosity as the outcome revealed similar effects, i.e., contradicting a hypothesis of true increasing effects (see Supplementary Results at https://osf.io/rk4ne/).

## Discussion

The objective of the present study was to illustrate how triangulation can be used to evaluate findings from RI-CLPM through reanalyses of data used by Ma and Wei [3]. Results from the original RI-CLPM (model 1) replicated Ma and Wei's finding of a statistically significant positive effect of initial within-individual residuals of curiosity on subsequent within-individual residuals of creativity when adjusting for initial within-individual residuals of creativity and vice versa. This finding was consistent with Ma and Wei's conclusion that curiosity and creativity mutually reinforce each other. However, five other models (models 2–6) suggested either null or a decreasing effect of curiosity on creativity and an additional model (model 7) suggested that data may have been generated without any direct effects between curiosity and creativity. The present divergent findings suggest that the data analyzed by Ma and Wei, and reanalyzed by us, offer no support to Ma and Wei's hypothesis that curiosity fosters creativity because it improves individuals' ability to develop original and useful ideas. Neither does the data appear to offer support to Ma and Wei's hypothesis, based on the information-gap theory [27], that creativity promotes curiosity because creativity prompts individuals to gather and scrutinize diverse information, make novel associations, gain fresh perspectives, and yearn for new knowledge.

Some researchers might think that a conclusion is valid as long as it is consistent with some information/data, even if the conclusion is contradicted by other information/data. According to such a viewpoint, Ma and Wei's conclusion that curiosity and creativity mutually reinforce each other would be perfectly valid even if contradicted by a majority of findings in the present study. These (hypothetical) researchers should, then, also think that it would be valid to conclude that P1 is the sought location in Fig 1, as this conclusion is consistent with the information that the sought location is 10 km from A, even if the conclusion disagrees with the information that the sought location is 7 km from B and 6 km from C. Moreover, these researchers should also think that it would be valid to conclude that curiosity has a decreasing effect on creativity in addition to concluding that curiosity has an increasing effect on creativity, as the present study found indications of both. We do not agree with this (hypothetical) position. Although empirical observations cannot always be taken at face value (due to small sample variance, measurement errors, etc.), as a general rule, conclusions should not be viewed as valid and trustworthy if they are contradicted by empirical observations, and conclusions claiming the simultaneous correctness of both X and its opposite should be avoided.

Another possible position could be that analyzed models should be rated according to their trustworthiness and then conclude whatever the most trustworthy model suggests. For example, let us assume that models 1–6 share 100 "trustworthiness-points" and that the original RI-CLPM (model 1) receives 20 points while models 2–6 receive 16 points each. In this case, the original RI-CLPM would be rated as the most trustworthy model and we would conclude, in agreement with Ma and Wei, that curiosity and creativity mutually reinforce each other. We do not agree with this (hypothetical) position either. As the sum of trustworthiness-points would be 20 for a hypothesis of reinforcing effects between curiosity and creativity and 80 against it, we do not think one should conclude that the hypothesis is correct.

The position we endorse is that conclusions should be based on an aggregation of all available evidence where, as far as possible, each piece of evidence is weighted according to trustworthiness. In the present case, without any strong opinions on how to rate trustworthiness of the various models, we could give them equal weight. Consequently we would, due to the ratio of contradicting to confirming findings, conclude that there is no strong evidence for reinforcing effects between curiosity and creativity in the present data. Of course, anyone giving more weight to the original RI-CLPM than to the other models combined could/should still conclude that curiosity and creativity mutually reinforce each other. However, we think that such an extreme valuation of the original RI-CLPM compared with the other models would require strong justification, especially when it is known that the RI-CLPM can give biased results [4–8].

We are aware that analyzing and considering results from several models is not the standard procedure in psychological research, where researchers usually pick what they assume/hope/are told is the best model for the analyses at hand. In the present case, anyone preferring the original RI-CLPM would probably, like Ma and Wei, conclude that curiosity and creativity mutually reinforce each other, while someone choosing the LCSM might claim the exact opposite. We think this would be unfortunate and speaks for the merit of analyzing the same data with several different models, i.e., to use triangulation. This corresponds to picking not just one of the three pieces of information in Fig 1, i.e., that the sought location is (i) 10 km from A; (ii) 7 km from B; (iii) 6 km from C, but, instead, to consider all of them when reaching a conclusion.

So far, we have only admitted to the conclusion that there was no strong evidence for reinforcing effects between curiosity and creativity in the data analyzed by Ma and Wei, and reanalyzed by us. Are we prepared to go any further? Are we, for example, prepared to claim a decreasing effect of curiosity on creativity as four models (models 2, 3, 5, and 6) suggested this, while just one model each suggested reinforcing (model 1) and null (model 4) effects. No, we are not prepared to claim a decreasing effect, because we know that a positive effect (indicating a decreasing effect) in models 2 and 5 is exactly what could be expected if curiosity and creativity were measured with less than perfect reliability (which should be very likely) and, therefore, susceptible to regression to the mean. Moreover, we also know that if measures of two constructs, e.g., curiosity and creativity, are affected by some common state factor (which also should be very likely), concurrent correlations ($r_{X1,Y1}$) will tend to be stronger than cross-lagged correlations ($r_{X1,Y2}$) which,

according to Equation 1 [28], would result in a negative effect of $X_1$ (e.g., initial curiosity) on the $Y_2$-$Y_1$ (e.g., subsequent – initial creativity) difference.

$$E(\beta_{X1,Y2-Y1}) = \frac{r_{X1,Y2} - r_{X1,Y1}}{\sqrt{2(1 - r_{Y1,Y2})}}$$

(1)

Instead, we propose that effects in models 1–6 could be accounted for by the model of spurious longitudinal associations (the MoSLA, model 7, Fig 3D). The MoSLA is, as a manner of speaking, P3 in Fig 1, consistent with all available information. However, the MoSLA was not apparently better than the other models, e.g., the original RI-CLPM, at reproducing variances of and covariances between observed variables. Therefore, we cannot claim that the MoSLA was superior to the original RI-CLPM based on model fit. However, unlike the RI-CLPM, or at least how it is commonly interpreted, the MoSLA does not make causal claims. In the present case, the MoSLA only suggested that a longitudinal association between curiosity and creativity may have been due to confounding by some trait common to curiosity and creativity and by common auto-correlated state factors, affecting curiosity and creativity measured at the same occasion. Consequently, unlike the RI-CLPM, or at least how the results were interpreted by Ma and Wei, the MoSLA was not falsified by results from the other models (i.e., models 2–6).

We believe that the present study, and the available analytic script (https://osf.io/rk4ne/), could be used as a model/tutorial by researchers wishing to scrutinize results from the RI-CLPM, i.e., when analyzing within-individual effects between two longitudinally measured constructs. The method of triangulation could be used with data on curiosity and creativity as well as observational (i.e., non-experimental) data on other constructs, both in psychology and other areas of research. Interested researchers are welcome to contact the corresponding author (KS) for advice and assistance. By fitting alternative models to data, predicting different signs of effects and using different measures of within-individual scores (person-mean centered scores vs. within-individual residuals in the RI-CLPM), researchers can seek validation for possible causal claims. If findings converge, claims can be made with increased confidence. If, on the other hand and as in the present study, findings diverge, caution is advised and strong claims should probably be avoided. The downside of closer scrutiny of one's findings might be a decreased likelihood for an opportunity to make strong and confident causal claims which, in turn, might decrease the likelihood for a publication [29–31]. However, we still believe, possibly somewhat controversially, that researchers should be quite sure of their case before making causal claims, and this should include, as far as possible, ruling out alternative explanations.

## Limitations

The present study suffered from some of the same limitations as the original study by Ma and Wei [3]. For example, all participants were employees at three firms in Hongkong, China, and it is unclear to what degree the present findings, that longitudinal associations between curiosity and creativity probably were spurious, generalizes to other cultural, social, and economical contexts. Hence, our findings may have been affected by cultural and sampling biases. However, we see no reason to believe that such biases would selectively have affected all models except the original RI-CLPM. Therefore, we do not believe that it would be tenable to argue that due to possible cultural and sampling biases, we should trust findings from the RI-CLPM, suggesting increasing prospective within-individual effects between curiosity and creativity, and dismiss contradictory findings from the other analyzed models.

The used measures, self-rated for curiosity and supervisor-rated for creativity, may not have been optimal and susceptible to measurement errors, bias, etc. However, it is important to bear in mind that the same data were used across analyzed models in the present study. Hence, possible shortcoming in measurements were constant across the analyzed models and could, consequently, not explain why the models indicated simultaneous and paradoxical increasing, decreasing, and null effects between curiosity and creativity.

## Conclusions

We used triangulation and reanalyzed data on curiosity and creativity used by Ma and Wei [3] with several models, e.g., the original random-intercept cross-lagged panel model (RI-CLPM), a latent change score model (LCSM), and multilevel regression analyses of person-mean centered scores. Only results from the original RI-CLPM were consistent with Ma and Wei's conclusion of mutually reinforcing effects between curiosity and creativity while results from the other models contradicted this conclusion. These contradictory findings and results from the model of spurious longitudinal associations (MoSLA) made us conclude that longitudinal associations between curiosity and creativity in the present data probably were spurious, possibly due to confounding by a trait common to curiosity and creativity and common auto-correlated state factors with effects on curiosity and creativity measured at the same occasion. The present study, and the available analytic script, could be used as a model/tutorial by researchers wishing to scrutinize results from the RI-CLPM, which is known to give biased results in some situations.

## Author contributions

**Conceptualization:** Kimmo Sorjonen, Bo Melin.

**Formal analysis:** Kimmo Sorjonen.

**Investigation:** Kimmo Sorjonen, Bo Melin.

**Methodology:** Kimmo Sorjonen, Bo Melin.

**Project administration:** Kimmo Sorjonen.

**Resources:** Bo Melin.

**Supervision:** Bo Melin.

**Validation:** Bo Melin.

**Visualization:** Kimmo Sorjonen.

**Writing – original draft:** Kimmo Sorjonen.

**Writing – review & editing:** Kimmo Sorjonen, Bo Melin.

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
