## [Decision Letter · Decision Letter 0]

1 Oct 2025

PONE-D-24-14717Using triangulation to evaluate findings from random-intercept cross-lagged panel models: An application with data on curiosity and creativityPLOS ONE

Dear Dr. Sorjonen,

Thank you for submitting your manuscript to PLOS ONE. After careful consideration, we feel that it has merit but does not fully meet PLOS ONE’s publication criteria as it currently stands. Therefore, we invite you to submit a revised version of the manuscript that addresses the points raised during the review process.

We look forward to receiving your revised manuscript.

Kind regards,

Vanessa Carels

Staff Editor

PLOS ONE

**Journal Requirements:**

1. When submitting your revision, we need you to address these additional requirements. Please ensure that your manuscript meets PLOS ONE's style requirements, including those for file naming. The PLOS ONE style templates can be found at https://journals.plos.org/plosone/s/file?id=wjVg/PLOSOne_formatting_sample_main_body.pdf and https://journals.plos.org/plosone/s/file?id=ba62/PLOSOne_formatting_sample_title_authors_affiliations.pdf 2. If the reviewer comments include a recommendation to cite specific previously published works, please review and evaluate these publications to determine whether they are relevant and should be cited. There is no requirement to cite these works unless the editor has indicated otherwise. 

Reviewers' comments:

Reviewer's Responses to Questions

**Comments to the Author**

1. Is the manuscript technically sound, and do the data support the conclusions?

Reviewer #1: Partly

Reviewer #2: Yes

Reviewer #3: Yes

2. Has the statistical analysis been performed appropriately and rigorously? 

Reviewer #1: Yes

Reviewer #2: Yes

Reviewer #3: Yes

3. Have the authors made all data underlying the findings in their manuscript fully available?

Reviewer #1: No

Reviewer #2: Yes

Reviewer #3: Yes

4. Is the manuscript presented in an intelligible fashion and written in standard English?

Reviewer #1: Yes

Reviewer #2: Yes

Reviewer #3: Yes

5. Review Comments to the Author

**Reviewer #1:**  1. General Assessment

The manuscript presents a relevant contribution to the literature by examining the applicability of the random-intercept cross-lagged panel model (RI-CLPM) in analysing the longitudinal relationship between curiosity and creativity. The study proposes critically examining this model, employing triangulation with different analytical approaches, such as latent change models and multilevel regression. The study's main conclusion challenges previous findings that suggest a bidirectional causal relationship between curiosity and creativity, arguing that such associations may be spurious due to common latent factors. Despite the topic's relevance, methodological and interpretation aspects of the results can be improved to strengthen the robustness of the conclusions.

2. Theoretical Foundation and Justification

The literature review presents a solid coverage of cross-lagged models, particularly the RI-CLPM, and their limitations in terms of causal inference. However, it could be expanded to include a more in-depth discussion on the suitability of the triangulation model in psychological research, exploring previous examples of its application. In addition, a greater alignment between the theoretical review and the formulation of hypotheses is recommended, making the relationship between the triangulated approaches and the gaps the study aims to fill more explicit.

3. Methodology

The choice of triangulation with multiple statistical models is a strength of the manuscript, as it increases the reliability of the findings. However, there is a need for greater clarity in the justification for selecting auxiliary models, such as the latent change model and multilevel regression with mean-centered scores. In addition, the description of the data used, which came from a sample of employees from three companies in Hong Kong, could be expanded to consider possible cultural and sampling biases that may impact the generalizability of the findings.

4. Analysis of the Results

The results indicate that only the RI-CLPM suggests significant cross-lagged effects between curiosity and creativity, while the other models provide contrary or null evidence. The introduction of the model of spurious longitudinal associations (MoSLA) as an alternative explanatory tool is an interesting innovation, but could be better supported. Furthermore, the interpretation of the findings could include a more detailed discussion of the possible impacts of variability in individual scores and the sensitivity of the models to the characteristics of the data used.

5. Discussion and Contribution

The discussion section correctly highlights the challenges in causal inference in longitudinal models and reinforces the importance of triangulated approaches. However, it would be beneficial to contextualize the findings within the psychological research field and apply these results in practice to future studies on creativity and curiosity. Furthermore, the suggested research agenda could include more specific recommendations on when and how triangulation should be used to analyze longitudinal data.

6. Conclusion and Recommendations

The manuscript represents a valuable contribution to understanding the limitations of cross-lagged models and promoting triangulation in longitudinal analyses. However, the following improvements are recommended before acceptance for publication:

- Expand the theoretical review to contextualize the use of triangulation in psychology better;

- Justify in more detail the choice of auxiliary models used in the analysis;

- Deepen the discussion on the implications of the findings for research on creativity and curiosity;

- Explore more clearly the limitations and potential biases of the sample used.

**Reviewer #2:**  This study re-examines prior findings that curiosity and creativity mutually reinforce each other using the random-intercept cross-lagged panel model (RI-CLPM). Acknowledging potential bias in RI-CLPM results, the authors apply triangulation with alternative methods, including a latent change score model (LCSM), multilevel regression with person-mean centering, and the novel Model of Spurious Longitudinal Associations (MoSLA). Contrary to the RI-CLPM, other models did not support mutual reinforcement, and MoSLA suggested the observed links may be spurious, possibly driven by shared traits or common state factors. The work underscores the need for model comparison in longitudinal research and offers analytic scripts as a resource for methodological scrutiny.

**Reviewer #3:**  In this paper, the authors reanalyzed data previously analyzed and used to draw a strong conclusion about the relationship between creativity and curiosity over time. Instead of using one model, these authors analyzed three models using both covariance (i.e., structural equation modeling) and mean-based (i.e., multilevel regression) methods as well as a fourth (i.e., seventh) model. Two of the models are similar expressions as the original data analysis. The version that matches the original analysis faithfully recreates the results, but no other models find the same results as the original analysis, and some find results that contradict the original analysis. The authors conclude that, on the weight of the analytical evidence, a strong association between creativity and curiosity across time cannot be supported.

This paper is clever and admirably succinct. The modeling choices are excellent, and the methods used are sufficient. I would love to see them take the leap into the Bayesian world, where comparing and integrating across models comes naturally, but that should not diminish this paper at all. The authors could have made different design decisions through the paper (and, indeed, I might have), but their design decisions are reasonable and lead to an excellent paper. I believe that this paper can be published as is. In case the authors would like things to work on, here are a few:

1) The weight of water example is good, but it can be very dense and non-intuitive. I imagine a lot of time was spent coming up with this example, and I do think it is good. Maybe a figure for this instead of the triangulation figure (which I think is much more intuitive for people) would be good. If not, maybe consider dividing that one big paragraph into smaller paragraphs and helping the reader move through it a little better. Insertion of connective tissue and more aids to the reader could help here.

2) Similarly, the first paragraph of the Results section is quite dense. Much of this is because of the repetition of "within-individual residual". As an applied statistician, I value the precision. As a reader, it makes it hard to focus on the things that actually matter. This might benefit from breaking it out into multiple paragraphs and really guiding the reader through it. Everything should be obvious. Maybe, since you've now spent so long getting them to understand the water example possibly even with a figure, you can tie everything very explicitly to the water example. In this water case, what would this finding imply? That might help people really see what fails.

3) It would be possible to reconceive of this paper as an explicit tutorial. I am not suggesting the authors do that but thinking of it that way may help them see where they might expand on a few points.

A minor point (but especially important if they want this to function as a tutorial): I believe that std.nox has been deprecated from the parameterEstimates function in lavaan. I was able to recreate the analysis anyway, but others may not be able to. You should make sure this works no matter what. Maybe use renv to create a virtual environment that ensures everyone gets the same versions of the packages and that it will work anywhere.

Note that I do not comment on writing directly, provide line edits, or copyedit in my reviews.

Curtis Atkisson

6. PLOS authors have the option to publish the peer review history of their article (what does this mean? ). If published, this will include your full peer review and any attached files.

**Do you want your identity to be public for this peer review?** For information about this choice, including consent withdrawal, please see our Privacy Policy .

Reviewer #1: No

Reviewer #2: No

Reviewer #3: **Yes: ** Curtis Atkisson

---

## [Author Response · Author response to Decision Letter 1]

8 Oct 2025

PONE-D-24-14717

Using triangulation to evaluate findings from random-intercept cross-lagged panel models: An application with data on curiosity and creativity

PLOS ONE

Dear Dr. Sorjonen,

Thank you for submitting your manuscript to PLOS ONE. After careful consideration, we feel that it has merit but does not fully meet PLOS ONE’s publication criteria as it currently stands. Therefore, we invite you to submit a revised version of the manuscript that addresses the points raised during the review process.

We look forward to receiving your revised manuscript.

Kind regards,

Vanessa Carels

Staff Editor

PLOS ONE

Reviewers' comments:

Reviewer's Responses to Questions

Comments to the Author

1. Is the manuscript technically sound, and do the data support the conclusions?

Reviewer #1: Partly

Reviewer #2: Yes

Reviewer #3: Yes

Response: We hope the amendments have increased the technical soundness of the manuscript.

2. Has the statistical analysis been performed appropriately and rigorously?

Reviewer #1: Yes

Reviewer #2: Yes

Reviewer #3: Yes

3. Have the authors made all data underlying the findings in their manuscript fully available?

Reviewer #1: No

Reviewer #2: Yes

Reviewer #3: Yes

Response: As we say in the manuscript, data used by Ma and Wei, and reanalyzed by us, is available at the Open Science Framework (https://osf.io/d48xa/) (line 135). Our analytic script, which downloads data used by Ma and Wei and then conducts all analyses reported in our manuscript, is available at the Open Science Framework (https://osf.io/rk4ne/) (line 205). Anyone can download our script, run it, and receive the same results as reported in our manuscript.

4. Is the manuscript presented in an intelligible fashion and written in standard English?

Reviewer #1: Yes

Reviewer #2: Yes

Reviewer #3: Yes

5. Review Comments to the Author

Reviewer #1:

1. General Assessment

The manuscript presents a relevant contribution to the literature by examining the applicability of the random-intercept cross-lagged panel model (RI-CLPM) in analysing the longitudinal relationship between curiosity and creativity. The study proposes critically examining this model, employing triangulation with different analytical approaches, such as latent change models and multilevel regression. The study's main conclusion challenges previous findings that suggest a bidirectional causal relationship between curiosity and creativity, arguing that such associations may be spurious due to common latent factors. Despite the topic's relevance, methodological and interpretation aspects of the results can be improved to strengthen the robustness of the conclusions.

Response: We hope our amendments have strengthened the robustness of the conclusions.

2. Theoretical Foundation and Justification

The literature review presents a solid coverage of cross-lagged models, particularly the RI-CLPM, and their limitations in terms of causal inference. However, it could be expanded to include a more in-depth discussion on the suitability of the triangulation model in psychological research, exploring previous examples of its application. In addition, a greater alignment between the theoretical review and the formulation of hypotheses is recommended, making the relationship between the triangulated approaches and the gaps the study aims to fill more explicit.

Response: We have added the following (lines 109-114):

We have previously used triangulation, i.e., analyzed data with alternative models, to challenge conclusions based on findings from analyses with the traditional CLPM [e.g., 9–11]. Moreover, in a recent study we used triangulation to scrutinize and challenge conclusions of causal effects of academic self-concept (i.e., self-perceived academic competence) on academic achievement based on findings from analyses with the RI-CLPM [12].

3. Methodology

The choice of triangulation with multiple statistical models is a strength of the manuscript, as it increases the reliability of the findings. However, there is a need for greater clarity in the justification for selecting auxiliary models, such as the latent change model and multilevel regression with mean-centered scores. In addition, the description of the data used, which came from a sample of employees from three companies in Hong Kong, could be expanded to consider possible cultural and sampling biases that may impact the generalizability of the findings.

Response: We have added the following (lines 193-202):

Analyzing models 2 and 5 were in line with proposals that time-reversed analyses may be used to identify potential statistical artifacts [19,20]. Previous analyses of data including an unquestionable causal effect (of adding stones in a container on the total weight of the container) found consistent effects in models 1-6 (i.e., effects in line with the predictions outlined above), suggesting that these models may be used to discriminate between true causal and spurious effects [21]. Model 7 (the MoSLA) was used in two recent studies and suggested that longitudinal data on academic self-concept and achievement [12] and on executive deficits and psychopathology [22], respectively, may have been generated without any direct effects between these constructs and, consequently, that previous causal claims could be challenged.

As we say in the manuscript (lines 393-396):

all participants were employees at three firms in Hongkong, China, and it is unclear to what degree the present findings, that longitudinal associations between curiosity and creativity probably were spurious, generalizes to other cultural, social, and economical contexts.

We have now added the following (lines 396-402):

Hence, our findings may have been affected by cultural and sampling biases. However, we see no reason to believe that such biases would selectively have affected all models except the original RI-CLPM. Therefore, we do not believe that it would be tenable to argue that due to possible cultural and sampling biases, we should trust findings from the RI-CLPM, suggesting increasing prospective within-individual effects between curiosity and creativity, and dismiss contradictory findings from the other analyzed models.

4. Analysis of the Results

The results indicate that only the RI-CLPM suggests significant cross-lagged effects between curiosity and creativity, while the other models provide contrary or null evidence. The introduction of the model of spurious longitudinal associations (MoSLA) as an alternative explanatory tool is an interesting innovation, but could be better supported. Furthermore, the interpretation of the findings could include a more detailed discussion of the possible impacts of variability in individual scores and the sensitivity of the models to the characteristics of the data used.

Response: We have added the following (lines 198-202):

Model 7 (the MoSLA) was used in two recent studies and suggested that longitudinal data on academic self-concept and achievement [12] and on executive deficits and psychopathology [22], respectively, may have been generated without any direct effects between these constructs and, consequently, that previous causal claims could be challenged.

Variability in individual scores is a prerequisite for associations and effects. Without variability, i.e., if all individuals would have the same score, it would not be possible to estimate associations or effects.

Moreover, as we say in the manuscript:

we know that a positive effect (indicating a decreasing effect) in models 2 and 5 is exactly what could be expected if curiosity and creativity were measured with less than perfect reliability (which should be very likely) and, therefore, susceptible to regression to the mean. Moreover, we also know that if measures of two constructs, e.g., curiosity and creativity, are affected by some common state factor (which also should be very likely), concurrent correlations (rX1,Y1) will tend to be stronger than cross-lagged correlations (rX1,Y2) which, according to Equation 1 [27], would result in a negative effect of X1 (e.g., initial curiosity) on the Y2-Y1 (e.g., subsequent - initial creativity) difference. (lines 348-356)

The used measures, self-rated for curiosity and supervisor-rated for creativity, may not have been optimal and susceptible to measurement errors, bias, etc. However, it is important to bear in mind that the same data were used across analyzed models in the present study. Hence, possible shortcoming in measurements were constant across the analyzed models and could, consequently, not explain why the models indicated simultaneous and paradoxical increasing, decreasing, and null effects between curiosity and creativity. (lines 403-408)

5. Discussion and Contribution

The discussion section correctly highlights the challenges in causal inference in longitudinal models and reinforces the importance of triangulated approaches. However, it would be beneficial to contextualize the findings within the psychological research field and apply these results in practice to future studies on creativity and curiosity. Furthermore, the suggested research agenda could include more specific recommendations on when and how triangulation should be used to analyze longitudinal data.

Response: We have added the following (lines 289-295):

The present divergent findings suggest that the data analyzed by Ma and Wei, and reanalyzed by us, offer no support to Ma and Wei’s hypothesis that curiosity fosters creativity because it improves individuals’ ability to develop original and useful ideas. Neither does the data appear to offer support to Ma and Wei’s hypothesis, based on the information-gap theory [27], that creativity promotes curiosity because creativity prompts individuals to gather and scrutinize diverse information, make novel associations, gain fresh perspectives, and yearn for new knowledge.

Moreover, as we say in the manuscript (lines 372-374):

We believe that the present study, and the available analytic script (https://osf.io/rk4ne/), could be used as a model/tutorial by researchers wishing to scrutinize results from the RI-CLPM,

We have now added the following (lines 374-378):

i.e., when analyzing within-individual effects between two longitudinally measured constructs. The method of triangulation could be used with data on curiosity and creativity as well as observational (i.e., non-experimental) data on other constructs, both in psychology and other areas of research. Interested researchers are welcome to contact the corresponding author (KS) for advice and assistance.

6. Conclusion and Recommendations

The manuscript represents a valuable contribution to understanding the limitations of cross-lagged models and promoting triangulation in longitudinal analyses. However, the following improvements are recommended before acceptance for publication:

- Expand the theoretical review to contextualize the use of triangulation in psychology better;

Response: We have added the following (lines 109-114):

We have previously used triangulation, i.e., analyzed data with alternative models, to challenge conclusions based on findings from analyses with the traditional CLPM [e.g., 9–11]. Moreover, in a recent study we used triangulation to scrutinize and challenge conclusions of causal effects of academic self-concept (i.e., self-perceived academic competence) on academic achievement based on findings from analyses with the RI-CLPM [12].

- Justify in more detail the choice of auxiliary models used in the analysis;

Response: We have added the following (lines 193-202):

Analyzing models 2 and 5 were in line with proposals that time-reversed analyses may be used to identify potential statistical artifacts [19,20]. Previous analyses of data including an unquestionable causal effect (of adding stones in a container on the total weight of the container) found consistent effects in models 1-6 (i.e., effects in line with the predictions outlined above), suggesting that these models may be used to discriminate between true causal and spurious effects [21]. Model 7 (the MoSLA) was used in two recent studies and suggested that longitudinal data on academic self-concept and achievement [12] and on executive deficits and psychopathology [22], respectively, may have been generated without any direct effects between these constructs and, consequently, that previous causal claims could be challenged.

- Deepen the discussion on the implications of the findings for research on creativity and curiosity;

Response: We have added the following (lines 289-295):

The present divergent findings suggest that the data analyzed by Ma and Wei, and reanalyzed by us, offer no support to Ma and Wei’s hypothesis that curiosity fosters creativity because it improves individuals’ ability to develop original and useful ideas. Neither does the data appear to offer support to Ma and Wei’s hypothesis, based on the information-gap theory [27], that creativity promotes curiosity because creativity prompts individuals to gather and scrutinize diverse information, make novel associations, gain fresh perspectives, and yearn for new knowledge.

- Explore more clearly the limitations and potential biases of the sample used.

Response: As we say in the manuscript (lines 393-396):

all participants were employees at three firms in Hongkong, China, and it is unclear to what degree the present findings, that longitudinal associations between curiosity and creativity probably were spurious, generalizes to other cultural, social, and economical contexts.

We have now added the following (lines 396-402):

Hence, our findings may have been affected by cultural and sampling biases. However, we see no reason to believe that such biases would selectively have affected all models except the original RI-CLPM. Therefore, we do not believe that it would be tenable to argue that due to possible cultural and sampling biases, we should trust findings from the RI-CLPM, suggesting increasing prospective within-individual effects between curiosity and creativity, and dismiss contradictory findings from the other analyzed models.

Reviewer #2:

This study re-examines prior findings that curiosity and creativity mutually reinforce each other using the random-intercept cross-lagged panel model (RI-CLPM). Acknowledging potential bias in RI-CLPM results, the authors apply triangulation with alternative methods, including a latent change score model (LCSM), multilevel regression with person-mean centering, and the novel Model of Spurious Longitudinal Associations (MoSL

---

## [Decision Letter · Decision Letter 1]

28 Oct 2025

PONE-D-24-14717R1Using triangulation to evaluate findings from random-intercept cross-lagged panel models: An application with data on curiosity and creativityPLOS ONE

Dear Dr. Sorjonen,

Thank you for submitting your manuscript to PLOS ONE. After careful consideration, we feel that it has merit but does not fully meet PLOS ONE’s publication criteria as it currently stands. Therefore, we invite you to submit a revised version of the manuscript that addresses the points raised during the review process.

The paper is poorly written; many points and claims are incomprehensible.

The triangulation problem and circle illustration in Fig. 1 make little sense and are disconnected from the statistical model used by the authors. For example, if the points are in a three-dimensional space, the author’s conclusion is wrong.  The authors do not solve a location problem, and therefore, this illustration does not work. I guess the authors meant that more information implies a more accurate answer or solution— a trivial statement.The statistical model, the heart of the discussion, is not defined. It is impossible to judge the validity of the conclusions and results until the model, along with all the involved variables depicted in Fig. 3, is mathematically defined in equation form (if the authors feel the model is too complicated, it may be presented in the Supplementary Materials file). The reader does not know what exactly the model is. The table with all estimated coefficients, along with standard errors and p-values, must be presented as well.

As a general critique, the paper lacks transparency regarding its modeling assumptions, making it difficult to assess the validity of the results.

We look forward to receiving your revised manuscript.

Kind regards,

Eugene Demidenko, Ph.D.

Academic Editor

PLOS ONE

Journal Requirements:

Additional Editor Comments:

The paper is poorly written; many points and claims are incomprehensible.

1. The triangulation problem and circle illustration in Fig. 1 make little sense and are disconnected from the statistical model used by the authors. For example, if the points are in a three-dimensional space, the author’s conclusion is wrong. The authors do not solve a location problem, and therefore, this illustration does not work. I guess the authors meant that more information implies a more accurate answer or solution— a trivial statement.

2. The statistical model, the heart of the discussion, is not defined. It is impossible to judge the validity of the conclusions and results until the model, along with all the involved variables depicted in Fig. 3, is mathematically defined in equation form (if the authors feel the model is too complicated, it may be presented in the Supplementary Materials file). The reader does not know what exactly the model is. The table with all estimated coefficients, along with standard errors and p-values, must be presented as well.

As a general critique, the paper lacks transparency regarding its modeling assumptions, making it difficult to assess the validity of the results.

Reviewers' comments:

Reviewer's Responses to Questions

**Comments to the Author**

1. If the authors have adequately addressed your comments raised in a previous round of review and you feel that this manuscript is now acceptable for publication, you may indicate that here to bypass the “Comments to the Author” section, enter your conflict of interest statement in the “Confidential to Editor” section, and submit your "Accept" recommendation.

Reviewer #3: All comments have been addressed

2. Is the manuscript technically sound, and do the data support the conclusions?

Reviewer #3: Yes

3. Has the statistical analysis been performed appropriately and rigorously? 

Reviewer #3: Yes

4. Have the authors made all data underlying the findings in their manuscript fully available?

Reviewer #3: Yes

5. Is the manuscript presented in an intelligible fashion and written in standard English?

Reviewer #3: Yes

6. Review Comments to the Author

Reviewer #3: This revision satisfactorily addresses all my comments. This paper represents a good reanalysis of existing data.

7. PLOS authors have the option to publish the peer review history of their article (what does this mean? ). If published, this will include your full peer review and any attached files.

**Do you want your identity to be public for this peer review?** For information about this choice, including consent withdrawal, please see our Privacy Policy .

Reviewer #3: **Yes: ** Curtis Atkisson

---

## [Author Response · Author response to Decision Letter 2]

30 Oct 2025

PONE-D-24-14717R1

Using triangulation to evaluate findings from random-intercept cross-lagged panel models: An application with data on curiosity and creativity

PLOS ONE

Dear Dr. Sorjonen,

Thank you for submitting your manuscript to PLOS ONE. After careful consideration, we feel that it has merit but does not fully meet PLOS ONE’s publication criteria as it currently stands. Therefore, we invite you to submit a revised version of the manuscript that addresses the points raised during the review process.

The paper is poorly written; many points and claims are incomprehensible.

1. The triangulation problem and circle illustration in Fig. 1 make little sense and are disconnected from the statistical model used by the authors. For example, if the points are in a three-dimensional space, the author’s conclusion is wrong. The authors do not solve a location problem, and therefore, this illustration does not work. I guess the authors meant that more information implies a more accurate answer or solution— a trivial statement.

Response: Although the point that more information allows more accurate answers may seem trivial, it appears not to be appreciated by many researchers. For example, it is quite common to conclude, as in the study by Ma and Wei (2023) that we challenge, causal effects based on cross-lagged effects without requiring any additional information. The point we are making with the triangulation example is that just as we should not infer a location based on its distance from another (known) location (something we assume/hope that all readers would agree with), we should not infer causality from a cross-lagged effect in observational data. We hope that this may open some eyes among researchers who have been lured into believing that cross-lagged panel models (with or without random intercepts) may, magically, reveal causality in observational data. We have added the following (line 41);

(in a two-dimensional space)

2. The statistical model, the heart of the discussion, is not defined. It is impossible to judge the validity of the conclusions and results until the model, along with all the involved variables depicted in Fig. 3, is mathematically defined in equation form (if the authors feel the model is too complicated, it may be presented in the Supplementary Materials file). The reader does not know what exactly the model is. The table with all estimated coefficients, along with standard errors and p-values, must be presented as well.

Response: We believe it to be standard practice to present structural equation models (SEM) as path diagrams rather than as equations. See, for example, Sorjonen et al. (2025) for a recent example by us and Toyoshima et al. (2020) for an example published in PLOS ONE. Such path-diagrams can also be found in very prestigious journals (e.g., Gosling et al., 2022). However, we have included documents with all effects in equation form as well as all output from the analyzed models (including standard errors and p-values) in our supplementary. We have added the following:

(regression equations for effects included in these models can be found in the document “Supplementary_Equations” available at https://osf.io/rk4ne/) (lines 148-150).

(full output for all models, including standard errors, p-values, etc., can be found in the document “Supplementary_Output” available at https://osf.io/rk4ne/) (lines 221-222).

As a general critique, the paper lacks transparency regarding its modeling assumptions, making it difficult to assess the validity of the results.

Response: We have made the analytic script publicly available (see line 216). Anyone with access to a computer and the internet can download the script, run it (with the statistical freeware R), and receive the same results as we did. This was confirmed in the previous round by reviewer # 3. Now we have also included documents with all effects in equation form as well as all output from the analyzed models (including standard errors and p-values) in our supplementary. We have added the following:

(regression equations for effects included in these models can be found in the document “Supplementary_Equations” available at https://osf.io/rk4ne/) (lines 148-150).

(full output for all models, including standard errors, p-values, etc., can be found in the document “Supplementary_Output” available at https://osf.io/rk4ne/) (lines 221-222).

We look forward to receiving your revised manuscript.

Kind regards,

Eugene Demidenko, Ph.D.

Academic Editor

PLOS ONE

Journal Requirements:

Additional Editor Comments:

The paper is poorly written; many points and claims are incomprehensible.

1. The triangulation problem and circle illustration in Fig. 1 make little sense and are disconnected from the statistical model used by the authors. For example, if the points are in a three-dimensional space, the author’s conclusion is wrong. The authors do not solve a location problem, and therefore, this illustration does not work. I guess the authors meant that more information implies a more accurate answer or solution— a trivial statement.

Response: Although the point that more information allows more accurate answers may seem trivial, it appears not to be appreciated by many researchers. For example, it is quite common to conclude, as in the study by Ma and Wei (2023) that we challenge, causal effects based on cross-lagged effects without requiring any additional information. The point we are making with the triangulation example is that just as we should not infer a location based on its distance from another (known) location (something we assume/hope that all readers would agree with), we should not infer causality from a cross-lagged effect in observational data. We hope that this may open some eyes among researchers who have been lured into believing that cross-lagged panel models (with or without random intercepts) may, magically, reveal causality in observational data. We have added the following (line 41);

(in a two-dimensional space)

2. The statistical model, the heart of the discussion, is not defined. It is impossible to judge the validity of the conclusions and results until the model, along with all the involved variables depicted in Fig. 3, is mathematically defined in equation form (if the authors feel the model is too complicated, it may be presented in the Supplementary Materials file). The reader does not know what exactly the model is. The table with all estimated coefficients, along with standard errors and p-values, must be presented as well.

Response: We believe it to be standard practice to present structural equation models (SEM) as path diagrams rather than as equations. See, for example, Sorjonen et al. (2025) for a recent example by us and Toyoshima et al. (2020) for an example published in PLOS ONE. Such path-diagrams can also be found in very prestigious journals (e.g., Gosling et al., 2022). However, we have included documents with all effects in equation form as well as all output from the analyzed models (including standard errors and p-values) in our supplementary. We have added the following:

(regression equations for effects included in these models can be found in the document “Supplementary_Equations” available at https://osf.io/rk4ne/) (lines 148-150).

(full output for all models, including standard errors, p-values, etc., can be found in the document “Supplementary_Output” available at https://osf.io/rk4ne/) (lines 221-222).

As a general critique, the paper lacks transparency regarding its modeling assumptions, making it difficult to assess the validity of the results.

Response: We have made the analytic script publicly available (see line 216). Anyone with access to a computer and the internet can download the script, run it (with the statistical freeware R), and receive the same results as we did. This was confirmed in the previous round by reviewer # 3. Now we have also included documents with all effects in equation form as well as all output from the analyzed models (including standard errors and p-values) in our supplementary. We have added the following:

(regression equations for effects included in these models can be found in the document “Supplementary_Equations” available at https://osf.io/rk4ne/) (lines 148-150).

(full output for all models, including standard errors, p-values, etc., can be found in the document “Supplementary_Output” available at https://osf.io/rk4ne/) (lines 221-222).

References

Gosling, W. D., Miller, C. S., Shanahan, T. M., Holden, P. B., Overpeck, J. T., & Van Langevelde, F. (2022). A stronger role for long-term moisture change than for CO2 in determining tropical woody vegetation change. Science, 376(6593), 653–656. https://doi.org/10.1126/science.abg4618

Ma, J. (Yonas), & Wei, W. (2023). Curiosity causes creativity? Revealing the reinforcement circle between state curiosity and creativity. The Journal of Creative Behavior, jocb.606. https://doi.org/10.1002/jocb.606

Sorjonen, K., Melin, B., & Nilsonne, G. (2025). Inconclusive evidence for a prospective effect of academic self-concept on achievement: A simulated reanalysis and comment on Marsh et al. (2024). Educational Psychology Review, 37(2), 30. https://doi.org/10.1007/s10648-025-10008-4

Toyoshima, K., Inoue, T., Masuya, J., Fujimura, Y., Higashi, S., Tanabe, H., & Kusumi, I. (2020). Structural equation modeling approach to explore the influence of childhood maltreatment in adults. PLOS ONE, 15(10), e0239820. https://doi.org/10.1371/journal.pone.0239820

Reviewers' comments:

Reviewer's Responses to Questions

Comments to the Author

1. If the authors have adequately addressed your comments raised in a previous round of review and you feel that this manuscript is now acceptable for publication, you may indicate that here to bypass the “Comments to the Author” section, enter your conflict of interest statement in the “Confidential to Editor” section, and submit your "Accept" recommendation.

Reviewer #3: All comments have been addressed

2. Is the manuscript technically sound, and do the data support the conclusions?

Reviewer #3: Yes

3. Has the statistical analysis been performed appropriately and rigorously?

Reviewer #3: Yes

4. Have the authors made all data underlying the findings in their manuscript fully available?

Reviewer #3: Yes

5. Is the manuscript presented in an intelligible fashion and written in standard English?

Reviewer #3: Yes

6. Review Comments to the Author

Reviewer #3: This revision satisfactorily addresses all my comments. This paper represents a good reanalysis of existing data.

7. PLOS authors have the option to publish the peer review history of their article (what does this mean?). If published, this will include your full peer review and any attached files.

Do you want your identity to be public for this peer review? For information about this choice, including consent withdrawal, please see our Privacy Policy.

Reviewer #3: Yes: Curtis Atkisson

---

## [Decision Letter · Decision Letter 2]

17 Nov 2025

Using triangulation to evaluate findings from random-intercept cross-lagged panel models: An application with data on curiosity and creativity

PONE-D-24-14717R2

Dear Dr. Sorjonen,

We’re pleased to inform you that your manuscript has been judged scientifically suitable for publication and will be formally accepted for publication once it meets all outstanding technical requirements.

Kind regards,

Najmul Hasan, PhD

Academic Editor

PLOS ONE

Additional Editor Comments (optional):

Reviewers' comments:

Reviewer's Responses to Questions

**Comments to the Author**

1. If the authors have adequately addressed your comments raised in a previous round of review and you feel that this manuscript is now acceptable for publication, you may indicate that here to bypass the “Comments to the Author” section, enter your conflict of interest statement in the “Confidential to Editor” section, and submit your "Accept" recommendation.

Reviewer #3: All comments have been addressed

2. Is the manuscript technically sound, and do the data support the conclusions?

Reviewer #3: Yes

3. Has the statistical analysis been performed appropriately and rigorously? 

Reviewer #3: Yes

4. Have the authors made all data underlying the findings in their manuscript fully available?

Reviewer #3: Yes

5. Is the manuscript presented in an intelligible fashion and written in standard English?

Reviewer #3: Yes

6. Review Comments to the Author

Reviewer #3: This revision satisfactorily addresses all my comments. This paper represents a good reanalysis of existing data. While the language may be dense at times, I think their changes make the paper understandable. I believe that the authors have been fully transparent in their presentation of the models. I believe this paper to be a good-faith effort to reanalyze existing data that successfully demonstrates that previous conclusions may not be warranted. This is a good contribution to the literature.

7. PLOS authors have the option to publish the peer review history of their article (what does this mean? ). If published, this will include your full peer review and any attached files.

**Do you want your identity to be public for this peer review?** For information about this choice, including consent withdrawal, please see our Privacy Policy .

Reviewer #3: **Yes: ** Curtis Atkisson

---

## [Editor Report · Acceptance letter]

PONE-D-24-14717R2

PLOS ONE

Dear Dr. Sorjonen,

I'm pleased to inform you that your manuscript has been deemed suitable for publication in PLOS ONE. Congratulations! Your manuscript is now being handed over to our production team.

Kind regards,

on behalf of

Dr. Najmul Hasan

Academic Editor

PLOS ONE